# The Relationship of Urbanization and Performance of Activity and Participation Functioning among Adults with Developmental Disabilities in Taiwan

**DOI:** 10.3390/ijerph17207553

**Published:** 2020-10-17

**Authors:** Shyang-Woei Lin, Tzu-Ying Chiu, Tsan-Hon Liou, Chia-Feng Yen, Hui-Guan Chen

**Affiliations:** 1Department of Natural Resources and Environmental Studies, National Dong Hwa University, Hualien 97401, Taiwan; shine@gms.ndhu.edu.tw; 2Graduate Institute of Long-term Care, Tzu Chi University of Science and Technology, Hualien City 97005, Taiwan; yin827@gmail.com; 3Department of Physical Medicine and Rehabilitation, Shuang Ho Hospital, Taipei Medical University, New Taipei City 23561, Taiwan; peter_liou@s.tmu.edu.tw; 4Department of Physical Medicine and Rehabilitation, School of Medicine, College of Medicine, Taipei Medical University, Taipei 11031, Taiwan; 5Department of Public Health, Tzu Chi University, Hualien City 97004, Taiwan; guanchen1023@gmail.com

**Keywords:** activity and participation functioning, developmental disabilities, performance, urbanization

## Abstract

Developmental disability is likely to be lifelong in nature and to result in substantial activity and societal participation limitations. The performance of individuals is related to the environment, supports, and urbanization of living cities. Most of the surveys for people with disabilities have not discussed the relationship between the cognitive impairment properties and performance of participation and activities functioning, and most cognitive impairments are regarded as having similar performance. The location of residence in childhood is mainly influenced by parents and main caregivers, but the factors related to the preferences of adults with cognitive impairment in the location of residence are more complicated. Objective(s): The aim was to explore and compare the relationships of the urbanization degree of their living cities and the functioning performance of daily living in various domains among adults with intellectual disability (ID), autism, and concomitant communicative impairment (CCI). Method: The cross-sectional study was applied, and the data was collected face-to-face by professionals in all authorized hospitals in Taiwan. The participants were 5374 adults with ID (n = 4455), autism (n = 670), CCI (n = 110) and combination disabilities (n = 139) which were according to the International Statistical Classification of Diseases 9th Revision (ICD-9) from a total of 167,069 adults with disabilities from the Disability Eligibility System (DES) in Taiwan between July 2012 and October 2013. The authors used the World Health Organization Disability Assessment Schedule 2.0–36 item version of WHO (WHODAS 2.0-36 items) to measure performance and capability of daily living. Results and Conclusions: There were significant differences in age, gender, disabled severity, and the urbanization between all subgroups (*p* < 0.05). After adjusting the age of all participators, the degree of urbanization just significantly affected the functioning score distribution in domain 1: cognition for an adult with ID, autism, and CCI; in domain 2, mobility for an adult with CCI and combination disability; in domain 3, self-care; domain 4, independent domains for ID (*p* < 0.05). There were no significant differences between urbanization degree and functioning scores in all domains for adults with autism. All in all, only in groups with combination disability did we find that the worse the degree of impairment was, the lower the degree of urbanization of their place of residence was, and there was no such phenomenon in adults with autism and ID in our study.

## 1. Introduction

Developmental disability is attributed mainly to health conditions or impairment of body functions, likely to be lifelong in nature and to result in substantial activity limitations and restrictions in societal participation. The performance of individuals is related to environment resources, supports, and urbanization of the living city [1].

According to the survey of Prevalence of Disability and Disability Types by Urban-Rural County Classification in the United States in 2016 [2], the number of adults with a disability was significantly higher in rural areas compared to large metropolitan areas. Compared with adults with disabilities living in urban areas, those in rural areas may face additional barriers, e.g., lower socioeconomic position, transportation problems, access to education and vocational rehabilitation services, access to health care and accessible communities for maintaining and improving their health, quality of life, and community participation [3]. Making rural communities disability inclusive and accessible can potentially improve the health and well-being of this population.

Compared with children, there are relatively few studies on adult developmental disorders. This study mainly focuses on the relationship between the three disability types of adults (intellectual disabilities (ID), autism, and concomitant communicative impairment (CCI) among developmental disorders) and urbanization of their living areas. Childrens’ living area mainly depends on parents’ choices, while an adult’s choice of place of residence is affected by many factors, including social welfare supports, the friendliness of the community, work/job status, and the adaptability of their capacity, etc. According to America Centers for Disease Control and Prevention’s (CDC’s) survey in 2016 [4], analyses were stratified by three age groups to find that among young adults (18–44 years old), compared with other disabilities like hearing, vision, cognition, mobility, self-care, and independent living, cognitive disability (10.6%) was the most prevalent type, which included intellectual disability, autism, and many mental and cognitive-relative diseases. They also found that persons with disabilities residing in the South U.S. Census region with lower socioeconomic status generally reported higher prevalence of disability [2]. The other survey from the American Community Survey (ACS) of the U.S. Census Bureau further analyzed the prevalence of various disability types between different degrees of urbanizations and found that average impairment rates were higher for micropolitan areas than metropolitan areas and were highest for noncore counties; this was consistent in six disability types [5,6].

Most of the prevalence surveys for people with disabilities have not discussed the relationship between the cognitive impairment properties and performance of participation and activity functioning. All cognitive impairments are regarded as a similar performance in most studies, but there are actually significant differences between people with cognitive and mental related diseases, intellectual disability, autism and language communication impairment. Clinically, most of the cognitive impairments caused by learning and language communication impairment in childhood require a large amount of language interaction and social stimulation, so a full support and resource environment is better for their development. For adults with intellectual disabilities, autism and living residence are rarely mentioned. There are many researches on the psychiatric and mental disorders and the geographical distributions focusing on geographically isolated, deinstitutionalized, and medical service accessibility issues [7,8,9,10,11,12,13]. The results of these studies are also consistent: the prevalence in rural areas was higher than urban areas, but studies still did not discuss the relationship between living area urbanization and the severity of the mental disorder or the performance of activity participation.

The location of residence in childhood is mainly influenced by parents and main caregivers, but the factors related to the preference of adults with cognitive impairment in location of residence are more complicated. This study wants to understand intellectual disability, autism, language communication and whether their activity and participation functioning in adulthood is related to their living urbanization situation.

This study assumes that there is a positive correlation between activity participation function and urbanization, but this relationship may not necessarily exist in every type of obstacle. Therefore, the purposes of this study are as follows: (1) To understand the differences in the performance of adults with intellectual disabilities, autism, language communication, and learning disabilities in adult participation; (2) to discuss the different relationships between the six domains of functioning, which are cognitive, moving, self-care, getting along with others, daily life and participation, and the urbanization of their place of residence.

## 2. Methods

This study was a cross-sectional study and employed secondary data analysis and drew data from the National Disability Eligibility Determination System (DEDS) in Taiwan between July 2012 and October 2013, a nationwide registry of the population with disabilities. The system contains the following information: basic demographic data, residence status (in institutions or in communities), impairment profile (e.g., the body function and body structure based on the International Classification of Health, Functioning, and Disability (ICF), functioning of daily living evaluation data which is based on the WHODAS 2.0-36 (World Health Organization Disability Assessment Schedule 2.0) 36 item version of WHO (including cognition, mobility, self-care, getting along, life activities and participation in six domains), and main ICD-9-CM codes of diseases. The impairment severity was parted into four classes: mild, moderate, severe, and profound, which were diagnosed by the physicians. The data were collected by 239 hospitals that were authorized to conduct disability evaluation in Taiwan. The evaluations were carried out by physicians and other professionals such as occupational therapists (OTs), physical therapists (PTs), speech therapists (STs), social workers, psychologists, and nurses.

The present study was approved by the Research Ethics Committee of the Hualien Tzu Chi Hospital, Buddhist Tzu Chi Medical Foundation (IRB102-178) and Joint Institutional Review Board Taipei Medical University (TMU-Joint Institutional Review Board), Taiwan. The functioning evaluation of the adults with disabilities was conducted using the Chinese version (in traditional Chinese) of the 36-item version of the World Health Organization Disability Assessment Schedule 2.0 (WHODAS 2.0-36 item) [14,15].

### 2.1. Participants

The participants were 5374 adults (above 18 years old) with ID (ICD-9-CM: 317-319), autism (ICD-9-CM: 299), and CCI (ICD-9-CM: 315.39, 748.5) who were also classified in body function and structure I and III based on the International Classification of Health, Functioning and Disability of World Health Organization from a total of 167,069 adults with disabilities who were officially registered in the DEDS in Taiwan from July 2012 to October 2013. The subgroups of the participants in the present study were 4455 adults with ID, 670 adults with autism, 110 adults with CCI and 139 adults with over two disorders of the above status (combination of ID, autism, and CCI).

### 2.2. Measures

The dependent variables in this study were the scores in six domains and summary functioning scores of WHODAS 2.0-36-items which evaluated the most recent 30-day experiences. The most important independence variable of “urbanization” was measured from socio-demographic variables which were developed by Taiwan Academia Sinica, which was monitored every five years. This investigation was from “Basic Survey of Social Changes in Taiwan” and promoted by the Humanities and Social Sciences Development Office of the National Science Council of the Executive Yuan in 1983 and planned and executed by researchers in the social sciences. The main purpose of the survey was to collect data through sample surveys to provide academics with research and analysis on social changes. In the design of the basic investigation and research, the principle of five-year intervals is the principle of conducting time-sensitive investigations, and the important goal of investigating social changes is to make a comparative analysis of more than two points in time. We collected the participants’ living villages and towns to define the urbanization of their living area based on the last version that is explained below.

#### 2.2.1. Functioning Score of WHODAS 2.0-36-Item

The WHODAS 2.0-36-item was developed based on the International Classification of Health, Functioning and Disability (ICF) of the WHO in 2010 to measure patients’ activities and participation in daily living in each of the following six domains within the previous 30 days: (1) cognition (6 items), by assessing communication and thinking activities such as concentrating, remembering, problem-solving, learning and communicating; (2) mobility (5 items), by assessing activities such as standing, moving around inside the home, getting out of the home and walking a long distance; (3) self-care (4 items), assessing activities such as hygiene, dressing, eating and staying alone; (4) getting along (5 items), by assessing interactions with other people and any difficulty experienced due to health conditions; (5) life activities (8 items), pertaining to the household, school, or work by assessing any difficulty experienced with day-to-day activities (activities that people perform on most days) which are associated with domestic responsibilities, leisure, work and school; and (6) participation (8 items), by assessing the social dimensions of the environment where the respondent resides such as community activities, barriers and hindrances, as well as problems encountered such as maintaining personal dignity. The possible responses to each item are 0: no difficulty, 1: mild difficulty, 2: moderate difficulty, 3: severe difficulty and 4: extreme difficulty, and all original scores in each domain would be transferred to 0–100 scores; the conversion process is based on the item response theory which was announced in WHODAS 2.0 [16]. WHODAS 2.0-36 items have been applied in many populations including patients with dementia, schizophrenia, Parkinson’s disease, cancers, and people with visual impairment and hearing impairment, etc. The WHODAS 2.0 covers six domains of life and is more comprehensive than other measurements like “Basic Activities of Daily Living (BADL)” and “Instrumental activities of daily living (IADL)” [17,18,19,20,21,22,23].

The domains and summary score ranges from 0 to 100 and the higher the score, the higher the level of disability. The participants answered 32 items if they were in non-employment status for the summary score, that is, the total of 36 items minus those related to employment and studying. The Chinese WHODAS 2.0-36-item was developed and published between 2013 and 2014 in Taiwan and has shown good validity and reliability [14]. The Table 1 shows the variables of questionnaire in this study.

#### 2.2.2. The Definition of the Urbanization

The urbanization was measured from city development and socio-demographic factors, including demographics, industrial and agricultural structures, commercialization, and public services, etc., which were highly related to various levels of development among boroughs and townships. The indexes of urbanization were population density, education level, and the status of economics that would affect the functioning of daily living among the people with developmental disabilities. The urbanization definition of the present study was according to factor analysis and cluster analysis by many sociologists and researchers to develop the Taiwan Academia Sinica, which monitors the factors that change every five years in Taiwan to confirm and modify the urbanization definition [24,25,26,27,28]. We collected the participants’ living villages and towns to define the urbanization degree of their living area based on the last version. There are seven degrees in our demographic statistics: metropolitan, tertiary industry city, new developed city, traditional industry city, less developed country, aging country, and periphery country, and we combined the aging with periphery country to do inferential statistics. The Figure 1 shows the six degrees of urbanization of our participants.

### 2.3. Data Analysis

Data were analyzed using the Statistical Package for the Social Sciences (version. 20.0, SPSS, Chicago, IL, USA), and the Geographic Information System (GIS, ArcGIS 10.3) to deal with the living location and urbanization variables. We used chi-square, ANOVA and T-test to test the difference between two variables. The application model and analysis strategy in this study were to find what difference between four sub-groups first, then to adjust the different factors to explore the relationship between functioning scores and the degree of urbanization of our participants’ living city or county. The index of urbanization was developed by many socio-demographic factors which included population distribution, employee status, age, and gender, etc., all in Table 2. In the present study, we only adjusted the age which is the key covariate both in their activity and participation functioning and urbanization.

To compare the different functioning scores between four subgroups after adjusting for age in varied urbanization, the authors used ANCOVA (Analysis of Covariance), Multivariate Statistical Analysis, Multivariate Analysis of Variance (MANOVA), and Multivariate Analysis of Covariance (MANCOVA). Before all of our analysis, we carried out homogeneity of variance. It was proved that the samples of each group were from the population with equal variance. We also used the Radar charts to illustrate the relationship between disability types, degree of urbanization, and functioning scores in various domains.

## 3. Results

### 3.1. The Characters of Our Participators

Table 2 shows the demographic characteristics of our participants, showing that adults with CCI had the oldest mean age (47.6 ± 16.3 years old), and the second was the subgroup with ID (30.7 ± 13.3). For gender, 88.1% of adults with autism, 79.1% in the combination impairments subgroup, 67.3% in the CCI subgroup, and 55.2% in the ID subgroup were male. The “mild” impairment severity was the highest percentage both in autism (63.1%) and CCI (54.5%) and “moderate” was the highest percentage in ID (41.6%) and combination impairments (39.6%). Over 90% participators among the ID, autism, and CCI subgroups were independent or half-dependent and lived in the community. About the urbanization, most of the adults with ID lived in the newly developed city (29.9%). For autism and the combination of impairments subgroups, most of them lived in the tertiary industry city (38.2%, 28.1%) and the subgroup of CCI mostly lived in the Metropolitan (29.1%). There were 94%–97% of participants living in the community among the ID, autism, and CCI subgroups. There were significant differences in age, gender, work status, impairment severity, urbanization of their living area, living status, and income of their households between all subgroups (*p* < 0.001) (Table 2).

To compare the performance of functioning score between these four groups, the highest summary score was in the combination of impairments subgroup (43.75 degrees) and the lowest summary score was in the autism subgroup (31.28 degrees). To compare the scores between six domains and four subgroups, we find that all in the mobility domain (Do2) and self-care domain (Do3) had lower scores, which means they had fewer limitations in mobility and self-care of daily life. The top two difficult domains were “Getting along” (Do4) and “Life activities of the household” (Do5-2) for our participants. The performance of each domain and summary functioning score were all significantly different between the four subgroups (*p* < 0.001) (Table 3).

### 3.2. The Relationship between Urbanization, Impairments, and Performance of Functioning

After adjusting the age of all participators (age was 29.88 years old among four subgroups), we found that the combination impairment subgroups had higher scores than other kinds of impairments and their performance was difficult in every domain no matter which level of urbanization (Table 4). Observing different domains of the status separately, we found that for the subgroup with ID, the urbanization level was only significantly difference in domain 2, domain 3 and domain 5-2 (*p* < 0.05). For the subgroup with autism, the urbanization level was not significantly different in any domain scores (*p* > 0.05). For the subgroup with CCI, the urbanization degree was only significantly different in domain 1, domain 4, and domain 5-2 (*p* < 0.05). For the subgroup with combination of impairments, the urbanization degree was just significantly difference in domain 1, domain2, domain 4, domain 6, and summary score (*p* < 0.05), and there was a consistent negative connection between the domain scores and urbanization degrees (the samples with higher scores lived in lower degrees of urbanization) within the combinational impairment subgroups. From the different aspects, in CCI we found that there were negative scores in domain 2, 3, 4, and 5-1 of the aging and the periphery county after adjusting for age. Their performance in mobility, self-care, getting along, and the work and school daily life was as fine in the aging and the periphery county as the general population.

Looking further from the radar chart (Figure 2), the relationship between the functional score of the groups and the degree of urbanization is directional. The hexagons of the radar map represent different urbanizations. There are four functional score lines in the map, representing four subgroups. Figure 2 shows the relationship of every domain score and urbanization among the four subgroups after adjusting age, and we modulated some negative scores to zero in the aging and the periphery county of the CCI subgroup. From Figure 2a–h, we found that in the aging and periphery country of every domain, the performance of the activity and participation functioning was the most restricted and difficult (with the highest mean score). In Figure 2f in the domain 5-2 life activities of work and school tasks, as the degree of urbanization increased, the score of all subgroups tended to decrease, but this phenomenon does not appear in the domain 2: mobility and domain 3: self-care (Figure 2b,c). In domain 6: participation (Figure 2g), we found that the scores among these six urbanizations were almost the same between adults with ID, autism, and CCI, except in the less developed countries.

## 4. Discussion

Our study results provided valuable evidence about the details on functioning status for six domains of daily life among the adults with different kinds of cognitive impairment, and to understand the relationship between the urbanization of their living setting and the functioning of the activity and participation. We found that in the four sub-groups of this study, there is a phenomenon of high urbanization in the distribution of their living cities, especially the subgroup with CCI whose proportion living in metropolitan cities is significantly higher than other types of cities, and other subgroups are mainly distributed in newly developed cities and tertiary industry cities. In Table 2, it can be found that the average age of people with CCI is 47.56, which is significantly higher than that of other sub-groups, so it is obviously not due to younger age; this is quite different than the previous survey of people with disabilities in the United States [2]. That could be indicating that the cities with high urbanization have better conditions for language communication technology aids, related environmental supports, or job opportunities than low-developed cities, but more evidence is still needed to clarify. It is worth noting that in the mean score in the Do3: Mobility, the adults with communication or language impairments are more restricted (higher score) in mobility than other types of obstacles, and their scores are even higher than those with multiple impairments (Table 3). That may be another reason for them to live in a place with a high degree of urbanization.

From the perspective of the relationship between the functional scores in Table 4 and the urbanization degree of the city in which they live, we found different results with the above paragraph of discussion in the Do1: Cognitive and Do2: Mobility among subgroups with communication and language impairments. After adjusting age, the cognitive score is indeed related to the degree of urbanization of the city in which they live, but it has nothing to do with Do2: Mobility. Their cognitive functioning scores are negatively correlated with the degree of urbanization. The lower the degree of urbanization, the higher the cognitive functioning scores (the worse the cognitive ability), and this phenomenon did not appear in people with ID and autism. Instead, the Do2: Mobility score of the people with ID is statistically significantly related to the degree of urbanization of the place of residence. Nevertheless, when we carefully observe the average scores of the Mobility domain of ID in different urbanizations, there is actually not a big gap between metropolitan, tertiary industry cities, and newly developed cities. Compared with the subgroups of combined disorders, we found that both in the cognitive domain and the mobility domain, the higher the scores of functioning, the lower the degree of urban development in the place where they live, which is similar to other research about psychosis [8,9,10]. The more complicated the impairment and the worse the functional performance, the more attention should be paid to the isolation situation of their living.

In addition, according to the results of this study, there is no significant difference between the functional scores of all the domains of adults with autism after adjusting age and the urbanization degree of their place of residence. It is shown that whether an adult with autism lives in metropolitan cities or remote aging areas, their environment does not have much to do with functional performance. This may imply two different meanings: (1) The urbanization development process in Taiwan, as far as the disease characteristics of adults with autism are concerned, the urban progress has already met the needs of autism, so there is no difference where they live; (2) On the other hand, it is a negative explanation; no matter what kind of urban development, for the environmental support needed for autistic adults, there is no special support or assistance. In the past, the results of research on autism on the choice of residence were not consistent. The research about the young autistic adults in the metropolitan city published in 2020 by Kaaren Haas et al. in Australian indicated that a triad of factors makes the use of public transport in a metropolitan city stressful for young autistic adults: (1) Their propensity to be intolerant of uncertainty; (2) the dominant role that anxiety plays; and (3) the impact of sensory processing, particularly the impact of crowding and associated tactile, auditory, and visual stimuli [13]. In the other research “Urbanicity and Autism Spectrum Disorders (ASD)” in 2014, they found a dose-response association with a greater level of urbanicity and risk of ASD prevalence and incidence rate. This association was found for residence at birth as well as a residence during childhood. Further, they found an increased risk of ASD in children who moved to a higher level of urbanicity after birth. Earlier age of ASD diagnosis in urban areas was observed. While they could not directly examine the specific reasons behind these associations, their results demonstrating particularly strong associations between ASD diagnosis and post-birth migration suggest the influence of identification-related factors, such as access to services, might have a substantive role on the ASD differentials we observed [29]. Obviously, the results of studies on the distribution of autism in adults and children are very different [30,31,32], and there is a lack of research on the relationship between a person with autism’s functional performance and urbanization. Even so, in the present study, there was not any association between their living cities and the functioning score for adults with autism. From the employment perspective, we also found that better functional performance of adults with ID and CCI was associated with higher urbanization cities. Areas with a higher degree of urbanization are better able to meet the needs of these two types of disabilities in terms of employment support and occupational diversity [17,21,33,34]. Particularly the analysis results of the subgroup with CCI had a more significant trend of people who lived in Metropolitan and Tertiary industry cities had the average functional score of its domain 5-2 at about 20 points.

## 5. Conclusions 

To conclude, from our research, only in groups with complex impairments combined with ID, autism, and CCI did we find that the worse the degree of impairment, the lower the degree of urbanization of their place of residence. This is consistent with the studies about patients with mental disorders often isolation in remote areas; there is no such phenomenon in adults with autism and ID in our study. That is to say, although ID, autism, and CCI all are with cognitive impairments, they have very different results in choosing a city to live in due to their different properties. It is not that there is a tendency to live in remote places as the disability level is more severe (Figure 2). According to the recommendations of The Global Network on Disability Inclusive and Accessible Urban Development (DIAUD), we are looking forward to urban development, which can provide people with various disabilities with more opportunities for social participation to achieve true equality of opportunity.

The biggest limitation of this study came from the inability to obtain more detailed urbanization or living setting variables from the secondary database, making the measurement variables supported by the environment relatively weak. The database is also cross-sectional data; it is not possible to explore the migration trajectory to understand the impairment and functional performance change and the relocation process of the residence. Lastly, the present study was an exploratory study and wanted to observe the social phenomena to understand whether our samples are geographically isolated and if the adults with different cognitive impairments have different relationships between the six domains of functioning and the urbanization degree. We still need an advanced study to develop or respond on a theoretical basis.

## Figures and Tables

**Figure 1 ijerph-17-07553-f001:**
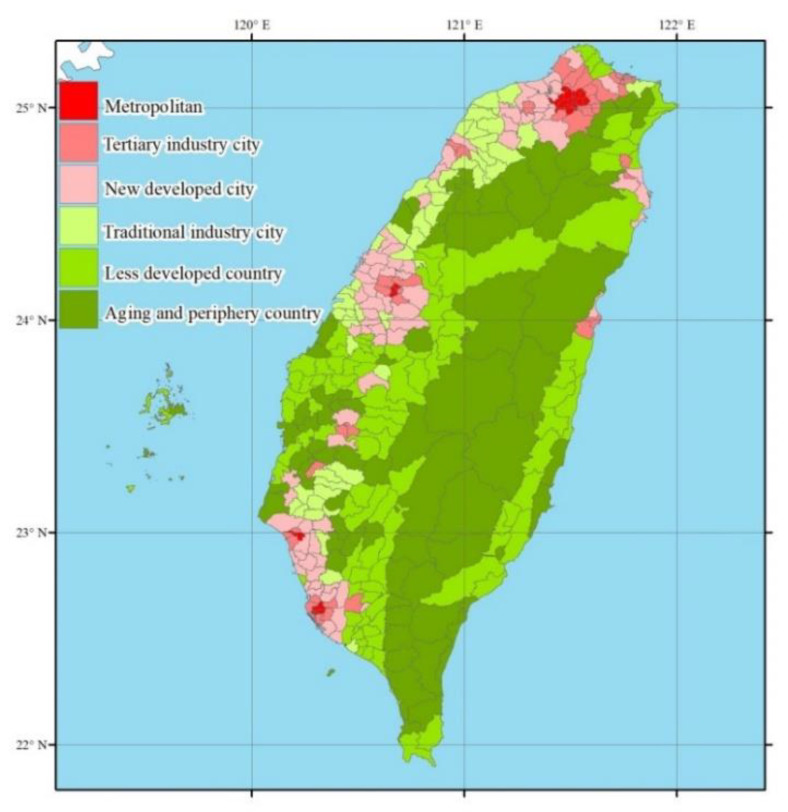
The Six Degrees Urbanization of our Participants.

**Figure 2 ijerph-17-07553-f002:**
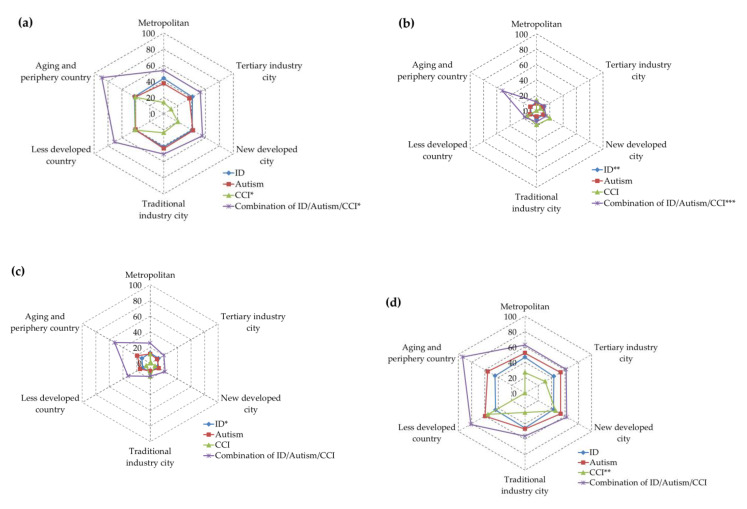
The performance functioning scores of WHODAS 2.0 between different urbanization: (**a**) Domain 1, Cognition; (**b**) Domain 2, Mobility; (**c**) Domain 3, Self-care; (**d**) Domain 4, Getting along; (**e**) Domain 5-1, Life activities: household; (**f**) Domain 5-2, Life activities: work and school task; (**g**) Domain 6, Participation and (**h**) Summary Scores. The adjusted scores significance between different urbanization and subgroups (*p* < 0.05), * *p* <0.05; ** *p* < 0.01; *** *p* < 0.001.

**Table 1 ijerph-17-07553-t001:** The variables and content of the questionnaire in our measurement.

Variables	Content	The Data Sources
1. Demography characters	Gender, birthday, the date of test, work status, living status, address of living, etc.	Self-report
2. Disease and Disability	ICD code, the type of disability, the severity of disability, etc.	To be diagnosed by the physicians
3. The activity and participation functioning	(1) Cognition (6 items), by assessing communication and thinking activities such as concentrating, remembering, problem-solving, learning and communicating; (2) Mobility (5 items), by assessing activities such as standing, moving around inside the home, getting out of the home and walking a long distance; (3) Self-care (4 items), assessing activities such as hygiene, dressing, eating and staying alone; (4) Getting along (5 items), by assessing interactions with other people and any difficulty experienced due to health conditions; (5) life activities (8 items-pertaining to the household, school, or work), by assessing any difficulty experienced with day-to-day activities (activities that people perform on most days) which are associated with domestic responsibilities, leisure, work and school; and (6) Participation (8 items),	The evaluations were carried out by physicians and other professionals such as occupational therapists (OTs), physical therapists (PTs), speech therapists (STs), social workers, psychologists, and nurses.

**Table 2 ijerph-17-07553-t002:** The characters of four subgroups (N = 5374).

Variables	ID n = 4455	Autism n = 670	Concomitant Communicative Impairment (CCI) n = 110	Combination of ID/Autism/CCI n = 139
**Age (Mean ± SD) *****	30.67 ± 13.27	23.19 ± 7.73	47.56 ± 16.32	22.66 ± 7.31
**Gender *****				
Male	2459 (55.2)	590 (88.1)	74 (67.3)	110 (79.1)
Female	1996 (44.8)	80 (11.9)	36 (32.7)	29 (20.9)
**Work status *****				
No	2640 (59.4)	262 (39.5)	49 (44.5)	82 (59.0)
Yes	965 (21.7)	99 (14.9)	37 (33.6)	18 (12.9)
Retire/students/house wife	836 (18.8)	302 (45.6)	24 (21.8)	39 (28.1)
**The severity of disability *****				
Mile	1281 (40.9)	423 (63.1)	60 (54.5)	17 (12.2)
Moderate	1853 (41.6)	159 (23.7)	22 (20.0)	55 (39.6)
Severe	592 (13.3)	66 (9.9)	9 (8.2)	40 (28.8)
Profound	189 (4.2)	22 (3.3)	19 (17.3)	27 (19.4)
**Urban Type of Living City (Urbanization) *****				
Metropolitan	724 (16.3)	199 (29.7)	32 (29.1)	33 (23.7)
Tertiary industry city	929 (20.9)	256 (38.2)	19 (17.3)	39 (28.1)
New developed city	1333 (29.9)	143 (21.3)	24 (21.8)	33 (23.7)
Traditional industry city	513 (11.5)	24 (3.6)	17 (15.5)	13 (9.4)
Less developed country	701 (15.7)	34 (5.1)	17 (15.5)	17 (12.2)
Aging country	182 (4.1)	9 (1.3)	0 (−)	4 (2.9)
periphery country	73 (1.6)	5 (0.7)	1 (0.9)	0 (−)
**Living status *****				
Living in community ^$^	4172 (94.4)	624 (94.3)	107 (97.3)	120 (86.3)
Living in an institution	249 (5.6)	38 (5.7)	3 (2.7)	19 (13.7)
**Income Status of Households ***** (n = 1657)				
Low income households	176 (12.3)	5 (3.4)	2 (7.4)	11 (22.9)
Middle-income households	97 (6.8)	4 (2.7)	0 (−)	1 (2.6)
General households	116 (85.5)	140 (94.0)	25 (92.6)	36 (75.0)

*** The variables were significantly different between the four subgroups that used the Chi-square (*p* < 0.001). ^$^ Includes independence or half-dependence living in the community.

**Table 3 ijerph-17-07553-t003:** The functioning scores means (Mean ± SD) of World Health Organization Disability Assessment Schedule (WHODAS) 2.0 between the four subgroups (n = 5374).

Domain	ID n = 4455	Autism n = 670	CCI n = 110	Combination of ID/Autism/CCI n = 139
Do1 Cognition ***	42.02 ± 25.39	34.96 ± 23.87	30.45 ± 29.89	52.63 ± 29.42
Do2 Mobility ***	10.84 ± 19.60	6.70 ± 14.43	22.27 ± 28.54	11.06 ± 18.57
Do3 Self-care ***	11.85 ± 19.23	9.69 ± 17.06	13.73 ± 25.80	21.94 ± 26.26
Do4 Get along ***	44.62 ± 29.67	49.56 ± 28.11	45.53 ± 33.42	60.91 ± 30.53
Do5-1 Life activities: Work and school task ***	40.32 ± 31.55	34.10 ± 31.37	32.09 ± 38.72	54.03 ± 35.50
Do5-2 Life activities: Household ***	56.31 ± 44.41	48.07 ± 41.16	45.13 ± 46.87	67.37 ± 39.91
Do6 Participation ***	31.84 ± 23.32	32.41 ± 23.06	38.41 ± 27.85	39.06 ± 26.60
Summary Scores ***	34.76 ± 20.54	31.28 ± 19.08	34.12 ± 25.70	43.75 ± 22.68

*** The scores of six domains were significant in four subgroups that used the ANOVA (*p* < 0.001).

**Table 4 ijerph-17-07553-t004:** The functioning scores of performances (mean ± SD) for all after adjusted age (covariance).

Variables: Age, Domain and Urbanization	ID n = 4455	Autism n = 670	CCI n = 110	Combination of ID/Autism/CCI n = 139
After adjust age	29.88			
Domain 1	^¥^ F = 1.38 *p* = 0.228	F = 1.345 *p* = 0.243	F = 2.814 * *p* = 0.020	F = 2.391 * *p* = 0.041
Metropolitan	43.65 ± 0.91	37.17 ± 1.74	13.68 ± 4.33	53.30 ± 4.25
Tertiary industry city	41.32 ± 0.80	37.02 ± 1.54	10.70 ± 5.62	52.76 ± 3.91
New developed city	40.93 ± 0.67	42.17 ± 2.05	20.39 ± 5.00	55.62 ± 4.25
Traditional industry city	41.63 ± 1.07	43.58 ± 4.98	23.98 ± 5.94	50.55 ± 6.77
Less developed country	40.89 ± 0.92	40.31 ± 4.18	41.16 ± 5.93	70.78 ± 14.08
Aging and periphery country	41.94 ± 1.53	40.85 ± 6.52	40.50 ± 24.39	88.94 ± 12.19
Domain 2	F = 3.028N ** *p* = 0.010	F = 0.259 *p* = 0.935	F = 0.843 *p* = 0.522	F = 4.420 *** *p* < 0.001
Metropolitan	11.71 ± 0.68	9.01 ± 1.31	13.85 ± 3.27	11.36 ± 3.20
Tertiary industry city	11.36 ± 0.60	9.76 ± 1.16	5.80 ± 4.23	11.34 ± 2.95
New developed city	10.36 ± 0.50	10.55 ± 1.54	20.01 ± 3.77	14.21 ± 3.20
Traditional industry city	11.02 ± 0.81	7.76 ± 3.76	18.65 ± 4.48	14.70 ± 5.10
Less developed country	8.41 ± 0.70	9.90 ± 3.15	14.48 ± 4.47	17.33 ± 4.46
Aging and periphery country	9.21 ± 1.15	9.40 ± 4.91	−1.32 ± 18.38 ^§^	51.71 ± 9.19
Domain 3	F = 2.514 * *p* = 0.028	F = 1.158 *p* = 0.329	F = 0.883 *p* = 0.495	F = 2.089 *p* = 0.071
Metropolitan	13.00 ± 0.70	11.54 ± 1.35	11.55 ± 3.37	25.72 ± 3.30
Tertiary industry city	12.48 ± 0.62	10.35 ± 1.19	1.09 ± 4.37	20.15 ± 3.04
New developed city	10.42 ± 0.52	12.85 ± 1.59	7.22 ± 3.88	22.01 ± 3.30
Traditional industry city	11.98 ± 0.84	10.05 ± 3.87	16.68 ± 4.61	16.58 ± 5.26
Less developed country	10.93 ± 0.72	14.80 ± 3.25	5.95 ± 4.60	32.68 ± 4.60
Aging and periphery country	12.09 ± 1.19	19.06 ± 5.01	−5.02 ± 18.95 ^§^	52.60 ± 9.47
Domain 4	F = 1.968 *p* = 0.08	F = 0.735 *p* = 0.597	F = 3.353 ** *p* = 0.008	F = 2.28 *p* = 0.05
Metropolitan	46.61 ± 1.07	52.07 ± 2.04	26.91 ± 5.09	62.71 ± 4.99
Tertiary industry city	43.30 ± 0.94	53.69 ± 1.80	30.51 ± 6.60	61.75 ± 4.59
New developed city	42.88 ± 0.79	53.80 ± 2.40	46.12 ± 5.87	62.74 ± 4.99
Traditional industry city	45.40 ± 1.27	46.62 ± 5.85	25.27 ± 7.00	55.94 ± 7.95
Less developed country	44.16 ± 1.08	60.00 ± 4.92	55.37 ± 6.96	81.34 ± 6.95
Aging and periphery country	44.80 ± 1.79	55.91 ± 7.66	−9.74 ± 28.65 ^§^	93.94 ± 14.32
Domain 5-1	F = 0.934 *p* = 0.458	F = 1.287 *p* = 0.268	F = 1.882 *p* = 0.104	F = 2.079 *p* = 0.072
Metropolitan	41.18 ± 1.15	34.65 ± 2.20	16.13 ± 5.48	55.57 ± 5.37
Tertiary industry city	39.28 ± 1.01	38.35 ± 1.94	10.68 ± 7.10	54.32 ± 4.94
New developed city	38.70 ± 0.85	40.24 ± 2.59	21.15 ± 6.31	62.36 ± 5.37
Traditional industry city	40.64 ± 1.36	39.97 ± 6.24	25.28 ± 7.50	41.83 ± 8.55
Less developed country	40.82 ± 1.16	43.84 ± 5.29	41.58 ± 7.49	69.00 ± 7.42
Aging and periphery country	39.75 ± 1.93	51.07 ± 8.23	−10.43 ± 30.81 ^§^	97.71 ± 15.40
Domain 5-2	F = 2.751 * *p* = 0.017	F = 1.135 *p* = 0.340	F = 3.085 * *p* = 0.012	F = 0.719 *p* = 0.610
Metropolitan	52.07 ± 1.59	52.30 ± 3.04	19.88 ± 7.29	78.99 ± 7.43
Tertiary industry city	53.87 ± 1.40	51.68 ± 2.68	21.44 ± 9.83	71.39 ± 6.84
New developed city	55.84 ± 1.17	56.59 ± 3.58	41.71 ± 8.74	69.07 ± 7.43
Traditional industry city	52.53 ± 1.88	46.46 ± 8.71	10.55 ± 10.38	75.80 ± 11.84
Less developed country	57.41 ± 1.61	59.02 ± 7.32	62.57 ± 10.36	66.39 ± 10.35
Aging and periphery country	60.80 ± 2.67	72.77 ± 11.40	85.91 ± 42.65	100.28 ± 21.32
Domain 6	F = 0.839 *p* = 0.522	F = 0.607 *p* = 0.694	F = 1.412 *p* = 0.226	F = 3.749 ** *p* = 0.003
Metropolitan	32.78 ± 0.84	34.59 ± 1.62	31.12 ± 4.03	39.83 ± 3.95
Tertiary industry city	31.18 ± 0.74	35.39 ± 1.43	20.54 ± 5.22	35.93 ± 3.64
New developed city	31.19 ± 0.62	37.45 ± 1.90	29.48 ± 4.65	43.02 ± 3.95
Traditional industry city	32.11 ± 1.00	32.25 ± 4.63	25.90 ± 5.52	41.27 ± 6.29
Less developed country	30.73 ± 0.86	34.02 ± 3.89	44.19 ± 5.51	52.42 ± 5.50
Aging and periphery country	30.88 ± 1.42	42.42 ± 6.06	25.23 ± 22.68	84.53 ± 11.34
Summary Scores	F = 0.961 *p* = 0.441	F = 1.018 *p* = 0.406	F = 2.086 *p* = 0.073	F = 3.228 ** *p* = 0.009
Metropolitan	35.29 ± 0.72	33.69 ± 1.37	20.91 ± 3.43	46.14 ± 3.36
Tertiary industry city	33.991 ± 0.63	34.34 ± 1.21	15.54 ± 4.44	43.39 ± 3.09
New developed city	33.77 ± 0.53	37.07 ± 1.62	27.00 ± 3.95	47.34 ± 3.36
Traditional industry city	35.25 ± 0.85	32.81 ± 3.94	22.20 ± 4.69	43.70 ± 5.35
Less developed country	33.96 ± 0.73	37.30 ± 3.31	39.23 ± 4.68	55.62 ± 4.67
Aging and periphery country	34.76 ± 1.21	41.37 ± 5.15	21.61 ± 19.27	81.79 ± 9.63

^¥^ ANCOVA results of four subgroups: After adjusting age in every subgroup, compared functioning scores between different urbanization. *, **, *** MANCOVA results: The adjusted score significance between different urbanization and subgroups * (*p* < 0.05), ** *p* < 0.01; *** *p* < 0.001 ^§^ The score was modified as 0 if the score was negative after adjusting for age in the following figure.

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
