# Peer review of "The Relationship of Urbanization and Performance of Activity and Participation Functioning among Adults with Developmental Disabilities in Taiwan"

_ijerph, 2020, doi:10.3390/ijerph17207553_

Round 1

Reviewer 1 Report

Review paper: “The Relationship of Urbanization and Performance of Activity and Participation Functioning among Adults with Developmental Disabilities in Taiwan” – International Journal of Environmental Research and Public Health.

In this research, The participants were 5,374 adults with ID (n=4455), autism (n=670), CCI (n=110), and combination disabilities (n=139) which were according to the ICD-9 from a total of 167,069 adults with disabilities from the Disability Eligibility System (DES) in Taiwan.

The research title is compelling and attractive and includes a promising good case study for Taiwan, especially the surveys for people with disabilities who have not discussed the relationship between the cognitive impairment properties and performance of participation and activities functioning, and most cognitive impairments are regarded as similar performance.

However, I have some suggestions for the author as follows:

  1. In the article, the author needs to set up a framework in order to have an overview of this application model, (especially which factors do affect to socio-demographic factor?)
  2. It is better to put a questionnaire into the table of content to convince readers
  3. In the line 28th -30th the author mentions: “The cross-sectional study was applied and the data was collected face to face by professionals in all authorized hospitals in Taiwan between July 2012 and October 2013”=> In the statistical research or scientific research, there is one negative point is that the research is only used for a short period of time, so whether this research data is still appropriate? Please explain more about this situation.
  4. Additionally, the research data is used from 2013 but the research target is from 2016, so the argument is inversely proportional.
  5. The author needs to prove that the liker 4 is appropriate and which previous studies have used this scale? Because as the table of contents is mentioned [16], this is a reporting analysis, not as scientific research.
  6. The author mentioned the Chi-square, ANOVA, and T-test analysis, however, the article does not show clearly the results of the analysis, and whether this is model is suitable?
  7. The author needs to explain more how the indicators do have meaning to the goal of the article. Example: F: how is the gain and P: how is suitable for the model?
  8. It is necessary to provide a theoretical basis to justify the results of the article, not based on subjective factors
  9. Why WHODAS 20-36 item is applied? Have any studies based on WHODAS? What difference does this article make for readers to see that it is appropriate for the research purpose?

Author Response

Dear Reviewer 

We would like to thank you for your valuable suggestions. We have revised the manuscript based on your suggestions and related to the full-text content in red color. Our responses and the pagination for the revisions in the revised manuscript are listed as follows with a point-by-point response to the reviewers’ comments. Please see the attached and the manuscript with re-submitted. (please see the section of the response to reviewer 1 in attached)

Reviewer 2 Report

The research questions and methodology are well-described with appropriate literature and evidence. The authors provide sufficient research context, information on the data sources, and analysis. Results are well-presented with appropriate discussion and conclusion. Overall, I would recommend this manuscript to be accepted after the following revisions/improvement:

(1) Since urbanization is a key factor in this study, the authors need to provide more information on 2.2.2 (The definition of urbanization). If possible, I suggest the manuscript include a map visualizing different areas based on the defined urbanization level.

(2) As Table 1 indicates, the sample size among sub-groups vary, and the ID sub-group has much larger samples (n=4455) comparing to the others. The authors need to further explain such sampling variation and potential impact on the modeling results.

(3) Some expressions cause confusion for the readers. For example, line 74 "different urbanizations", line "living urbanization situation", line 184 "degree of urbanization", etc. Are these terms indicating the same measure, which is the urbanization categorical variable? If so, I suggest the author improve the consistency of expression and terminology.

(4) This manuscript provides a clear data visualization in Figure 1. Between line 234 and 236, the authors need to add one sentence to explain how to read the spider plots and the interpretations in this research context.

Minor remarks: Some typos and expression variations need to be fixed, for example, U.S. vs. US.

Author Response

Dear Reviewer 

We would like to thank you for your valuable suggestions. We have revised the manuscript based on your suggestions and related to the full-text content in red color. Our responses and the pagination for the revisions in the revised manuscript are listed as follows with a point-by-point response to the reviewers’ comments. Please see the attached and the manuscript with re-submitted. (please see the section of the response to reviewer 2 in attached)
